# Control of Surface Plasmon Resonance in Silver Nanocubes by CEP-Locked Laser Pulse

**Ju Liu [1] and Zhiyuan Li [2],***

[1]  School of Science, China University of Mining and Technology-Beijing, Beijing 100083, China; juliu67@sina.com

[2]  College of Physics and Optoelectronics, South China University of Technology, Guangzhou 510641, China

*  Correspondence: phzyli@scut.edu.cn

**Abstract:** Localized surface plasmon resonance (LSPR) of metal nanoparticles has attracted increasing attention in surface-enhanced Raman scattering, chemical and biological sensing applications. In this article, we calculate the optical extinction spectra of a silver nanocube driven by an ultrashort carrier envelope phase (CEP)-locked laser pulse. Five LSPR modes are clearly excited in the optical spectra. We analyze the physical origin of each mode from the charge distribution on different parts of the cubic particle and the dipole and quadrupole excitation features at the LSPR peaks. The charge distribution follows a simple rule that when the charge concentrates from the face to the corners of the cubic particle, the resonant wavelength red-shifts. Then we modulate the LSPR spectra by changing CEP. The results show that CEP has selective plasmon mode excitation functionality and can act as a novel modulation role on LSPR modes. Our work suggests a novel means to regulate LSPR modes and the corresponding optical properties of metal nanoparticles via various freedoms of controlled optical field, which can be useful for optimized applications in chemical and biological sensors, single molecule detection, and so on.

**Keywords:** surface plasmons; ultrashort laser pulse; carrier envelope phase; nanoparticles; nanocubes; modulation

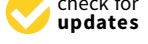



## 1. Introduction

The optical properties of metal nanoparticles have long been of great interest in physics, chemistry, biology and their interdisciplinary fields. Localized surface plasmon resonance (LSPR), which is a collective oscillation of conduction electrons, occurs when metal nanoparticles interact with incident light waves [1]. This resonance leads to large enhancements of local electromagnetic field around the nanoparticle surface [2] and is sensitive to nanoparticle size, shape, composition and deposited substrate as well as the external dielectric environment [3]. These unique properties make the research of LSPR spectroscopy significant for application to surface-enhanced Raman scattering (SERS) [4], chemical and biological sensing [5], antennae [6,7] and so on.

For many metal nanoparticles, the optical extinction spectra present a series of LSPR peaks in a broad spectral range with different amplitudes and widths. Understanding and predicting the fundamental physics governing LSPRs is both necessary to realize and fully optimize potential devices. Mie theory can precisely calculate the LSPR modes of a spherical Ag nanoparticle with size larger than 80 nm. Higher plasmon modes appear in the optical spectrum and different modes would localize at different spatial regions of the nanoparticle [8–10]. In comparison, the research on Ag nanocubes is much less complete, as an analytical solution is not available for precise calculation of their optical spectra. Fuchs was the first to envision theoretically the response of a cubic particle described by a model dielectric constant in the electrostatic limit and predicted several surface phonon modes of ionic nanocubes [11,12]. When a cubic particle interacts with continuous wave (CW) light, one or two LSPR peaks are observed for smaller particles with sizes below

80 nm, and a third peak appears between the two peaks when the nanocube edge length is longer than 90 nm [1]. However, these LSPR peaks, including both the width and amplitude, are difficult to distinguish clearly and precisely. As a result, previous studies have mainly concentrated on analysis of the main dipole and quadrupole modes while ignoring higher-order modes [13,14].

To solve the problem and clearly identify all plasmon modes in metal nanoparticles, we consider the interaction of a nanocubic Ag nanoparticle with an ultrashort laser pulse. The emergence of ultrashort pulse laser technology [15,16] has provided a powerful tool for probing physical problems in an unprecedented fast timescale. Recent progress in ultrashort pulse laser technology has resulted in the generation of intense optical pulses comprising only a few wave cycles within the full width at half maximum (FWHM) of their temporal intensity profile [17–21]. The study of the interaction of intense few-cycle laser pulses with matter has brought a new, important branch of investigations in nonlinear optics [22] and has opened up a number of applications ranging from nanometer-scale materials processing [23] to the generation of coherent soft-X-ray radiation for biological microscopy [24]. With the FWHM of laser pulse becoming comparable to the time period of oscillation cycles, the temporal evolution of the electric and magnetic fields of a few-cycle light pulse and, hence, all nonlinear processes driven by these fields become increasingly affected by the carrier-envelope phase (CEP) of the pulse [25–28]. For instance, in 2008 Goulielmakis and coworkers measured the sub-femtosecond XUV emission from neon atoms ionized by a linearly polarized, sub-1.5-cycle, 720-nm laser field [29]. The ratio of the energy of the main attosecond XUV pulse to the overall XUV emission energy transmitting through the bandpass strongly depends on the CEP. At present, the regulation and control of LSPR is made mainly by selecting the particle size and shape, as well as the surrounding dielectric medium [30,31]. The emergence of ultrashort pulses hopefully may provide another regulative parameter, due to the CEP effect.

In this paper, we calculate the optical extinction spectra of a silver nanocubic particle driven by an ultrashort CEP-locked pulse. We analyze the physical origin of the five LSPR modes appearing in the spectra by examining the distribution patterns of electric fields and electric charges on the face of the cube. The result shows that when the charge concentrates from the face to the corners of the cubic particle, the resonant wavelength redshifts. We then investigate the influence of the CEP of laser pulse on the regulation and control of the LSPR modes.

## 2. Materials and Methods

The optical extinction spectra of a silver nanocubic particle is calculated by three-dimensional (3D) finite-difference time-domain (FDTD) simulations [32,33]. For comparison, we consider the interaction of a 3D silver nanocubic particle of edge length 90 nm with a CW light and an ultrashort CEP-locked pulse. The particle, as schematically depicted in Figure 1a, is embedded within an air background with a refractive index of 1.0. We choose the center of the silver particle as the origin of coordinates, and the edges of the nanocube are parallel to the $x$, $y$ and $z$ axes. The incident light propagates along the $z$ axis and the polarization direction is along the $x$ axis. We adopt the optical constants given in Palik [34] and calculate the extinction cross section of Ag nanoparticle by using the 3D FDTD method. In our calculation, a large simulation span of 3 um and a long simulation time of 800 fs are used. The simulation region uses PML absorbing boundary conditions on all boundaries and the maximum mesh step is 2 nm. To calculate the extinction spectra, we set an analysis group located inside the source to measure the absorption cross-section and an analysis group located outside the source to measure the scattering cross-section. Meanwhile, a two-dimensional monitor is set in the $xz$ plane to calculate the electric field intensity and an analysis group with span of 100 nm to calculate the charge density distribution.

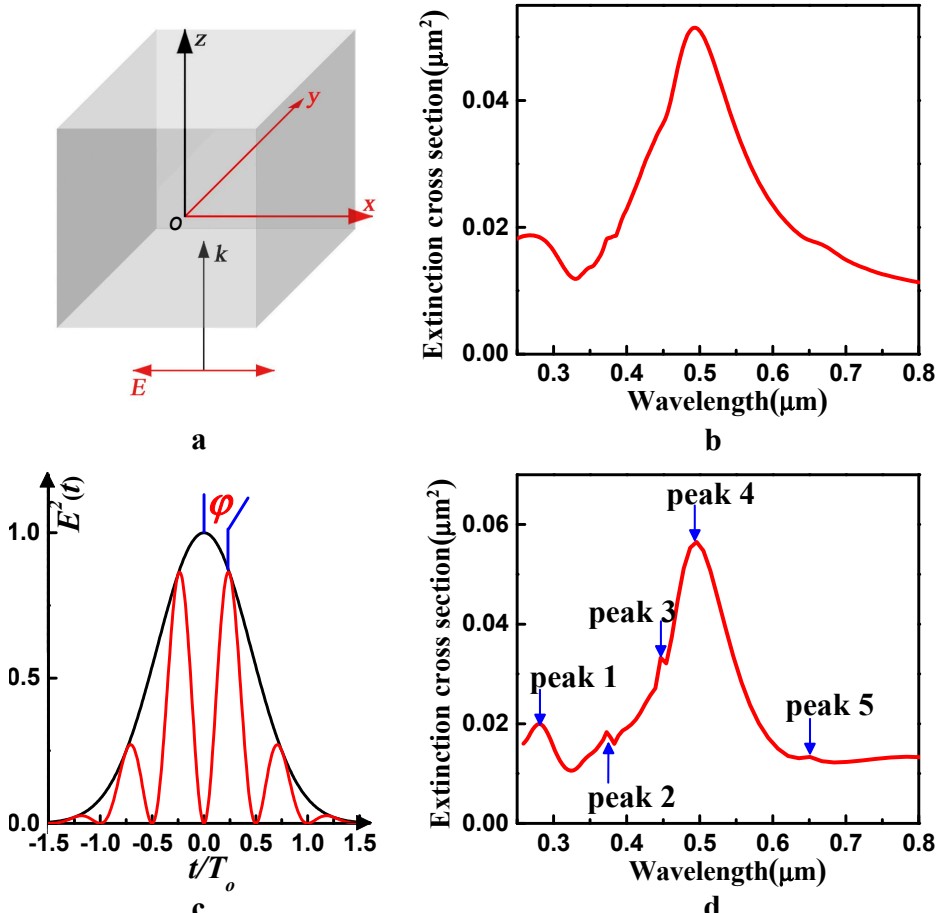

**Figure 1.** (**a**) Schematic configuration of the nanocube and the coordinate system for the simulation. (**b**) Calculated optical extinction cross section spectra for the Ag nanocube embedded in an air background with an edge length of 90 nm driven by a continuous wave light. (**c**) Schematic of an ultrashort pulse. $\varphi$ is the CEP, $T_0$ is the time period of the laser field. (**d**) Calculated optical extinction cross section spectra for the Ag nanocube embedded in an air background with an edge length of 90 nm driven by an ultrashort pulse.

## 3. Results

We first consider the interaction of a silver nanocubic particle with a CW light. The result is shown in Figure 1b. We find that only one dominant LSPR peak located at 494 nm can be observed obviously in the spectrum, whereas other peaks are hardly recognizable and may overlap with each other. Previous studies focus on analysis of the contributions of the dipole component and quadrupole component of the dominant peak to the optical cross sections, while much less attention has been paid to the other plasmon modes when driven by a CW light.

To solve the problem and figure out the plasmon modes of the silver particle clearly, we employ a CEP-locked ultrashort pulse to interact with the silver nanocubic particle, as illustrated in Figure 1c. The electric field of the laser pulse is written as

$$E(t) = E_0 \exp(-\alpha^2 t^2) \cos(\omega t + \varphi) \tag{1}$$

where $E_0$ is the amplitude, $\alpha = \sqrt{2\ln(2)}/\tau$, with $\tau$ being the FWHM of the pulse, $\omega$ and $T_0$ are the frequency and time period of the laser field, respectively, and $\varphi$ is the CEP. In the calculation, the amplitude is set as 1 V/m and $\varphi = \pi/3$. The FWHM duration of the laser pulse is one optical cycle, and the carrier frequency of the laser field is $4.7 \times 10^{15}$ Hz , which corresponds to 0.11388 atomic units (a.u.). As shown in Figure 1d, we clearly observe five LSPR peaks ordered as peaks 1–5 for this 90 nm silver cube, and they are located at

275 nm, 374 nm, 447 nm, 495 nm and 643 nm, respectively. These peaks correspond to five kinds of plasmon modes. The dominant peak is peak 4, which is located at 495 nm. The other peaks are much weaker than peak 4. Therefore, the overall extinction spectrum curve looks like a broad peak centered at 495 nm with three shoulders appearing on the short-wavelength side and one shoulder appearing on the long-wavelength side. Here, we try to assign unambiguously each LSPR peak to the excitation of either a dipole, or quadrupole, or hybrid resonance mode. Furthermore, to better understand the intrinsic nature of these plasmon resonance peaks, we analyze the physical origin of different LSPR peaks essentially from the surface charge distribution.

## 4. Discussion

The local field distribution has been widely utilized to discern the physical nature of a particular plasmon resonance mode appearing in the extinction spectrum curve. In fact, the optical properties of a particle are associated with surface modes, which are accompanied by polarization charge on the surface [11]. In other words, the optical extinction of the particle can be associated with the surface polarization charge. To understand clearly the physical origin of the surface modes, we calculate the electric field intensity patterns and the surface charge distributions for a silver particle under different incident waves. In the field intensity calculations, we have selected an enclosed cubic surface with a 2 nm gap from the surface of the nanocube. Figure 2 shows the calculated electric field intensity (in units of the incident field intensity throughout this paper) and charge density distribution (in units of $C/m^3$ throughout this paper) of the 90 nm silver nanocube in the $xz$ plane with $y = -45$ nm at the 275 nm, 374 nm, 447 nm, 495 nm and 635 nm resonance peaks, respectively. Figure 2a shows the local electric field intensity distribution at the wavelength of 275 nm, whereas Figure 2b illustrates the corresponding polarized electric charge distribution. From the figures, we can clearly see that mode 1 is a dipole mode because charges of different sign are separated on the left and right parts of the face of the cube. For mode 2 and mode 3, the charge distribution shown in Figure 2d,f illustrates that both the 374 nm and 447 nm resonance modes are quadrupole modes, as the signs of the electric charge on the two $z$-axis edges are opposite to each other, and they are also opposite on the bottom side and upper side on one $z$-axis edge.

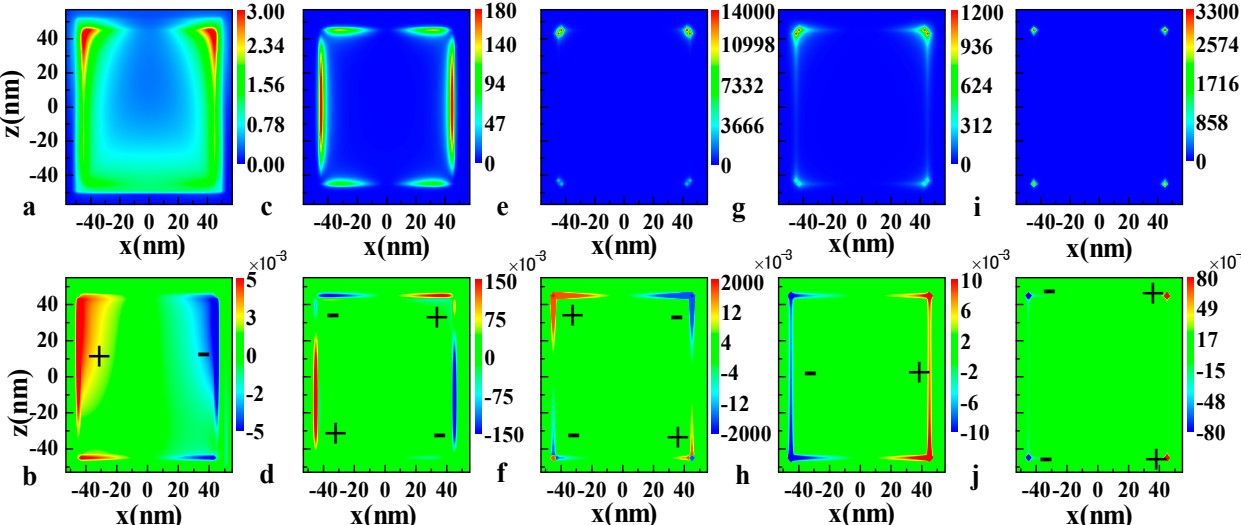

**Figure 2.** Calculated near-field intensity in the $xz$ plane of a 90 nm silver nanocube at the (**a**) 275 nm, (**c**) 374 nm, (**e**) 447 nm, (**g**) 495 nm and (**i**) 643 nm resonance peaks, respectively. The corresponding polarized electric charge distribution at (**b**) 275 nm, (**d**) 374 nm, (**f**) 447 nm, (**h**) 495 nm and (**j**) 643 nm resonance peaks, respectively. Red and blue colors indicate positive and negative charge, which are also indicated by "+" and "−" marks, respectively.

According to Figure 2c,e, the field intensity is high at the edge of the cube for the 374 nm resonance mode, but for the 447 nm resonance mode the intensity is more concentrated in the corners. For mode 1 and mode 2, the polarization charges cover the face and the edges of the cube, respectively. However, polarization charges tend to concentrate in the corners when the resonance wavelength redshifts. Taking a closer look at Figure 2e,f, we find that both the local field and the charge distribution are larger on the upper side of the nanocube than on the bottom side. The difference is attributed to the retardation effect of the incident light, which becomes significant when the size of the particle increases to 90 nm. On the other hand, the charges in the two sharp corners at the bottom side are not strictly opposite in sign when compared with the upper side. Therefore, the excited plasmon mode at this wavelength is not a pure quadrupole mode. It can be assumed to be the superposition of dipole and quadrupole modes, and the quadrupole mode is dominant here. Figure 2g,h clearly show that the 495 nm mode corresponds to a dipole mode induced by the $x$ axis polarized incident waves as the electric charge located on the edges and corners has opposite sign on the two $z$-axis edges in the $xz$ plane with $y = -45$ nm. The intensity is concentrated in a larger volume near the upper two corners. Finally, mode 5, namely, the 635 nm resonance mode, is also a dipole mode and the charge is highly localized in the region that is very close to the corners as shown in Figure 2i,j.

From the discussion above, we can find several interesting features. First, the charge distribution follows a certain rule. Generally, the charges distributed on the faces of a cube contribute to the short-wavelength resonance mode while charges localized at the corners correspond to the long-wavelength surface mode. The mode generated by the edges of a cube locates between the two modes, and some modes originate from the combination of the faces, edges and corners. According to Ref. [11], the polarization charges on different parts of a nanocube produce different normal components of the polarization at the surface and result in the difference of dipole moment, which directly determine the surface mode frequencies. The relationship between the resonance frequencies and the charge distribution can be written in the form:

$$\langle \chi(\omega) \rangle = \sum \frac{C_m}{\chi^{-1}(\omega) + 4\pi n_m}, \tag{2}$$

where $\chi(\omega)$ is the complex dielectric susceptibility of the particle, $n_m$ is the depolarization factor associated with the $m$th normal mode and $C_m$ denotes the strength of the mode. A resonant peak occurs when $Re[\chi^{-1}(\omega)] \approx -4\pi n_m$ or $\varepsilon'(\omega) = 1 - n_m^{-1}$, where $\varepsilon'(\omega)$ is the value of the real part of the material dielectric function at each resonant peak. The depolarization factor decreases when the charge concentrates from the face to the corners of the cubic particle. As a result, the value of the real part of the material dielectric constant requested to support plasmon resonance decreases, and this corresponds to the redshift of the resonant wavelength. Second, as the particle is as large as 90 nm, quadrupole modes are excited by the incident wave at the metal nanoparticle due to the phase retardation of the field inside the particle. The 275 nm, 495 nm and 643 nm peaks are induced by the dipole resonances, whereas the 374 nm and 447 nm resonance modes are mainly quadrupole modes. The difference in the charge distribution and the contributions of the dipole component and quadrupole component reveal the physical origin of the five plasmon modes.

Third, the field enhancement factor does not scale proportionally to the extinction efficiency. Figure 1b shows that the dominant peak of the extinction cross section of a 90 nm Ag nanocube is 495 nm resonance mode while the 447 nm resonance mode is a shoulder of the dominant mode. Whereas the field intensity calculated in Figure 2e,g indicates that at a wavelength of 447 nm, the maximum field enhancement factor is 14,590, while it is only 1200 at a wavelength of 495 nm. Therefore, the 447 nm mode is a large field enhancement and low extinction (LFE-LE) resonance mode, according to the analysis made by Zhou and coworkers [35]. The physical mechanism behind this peculiar feature is that the polarization charge is highly densified in a very limited volume around the corners of the nanocube. As

shown in Figure 2e,g, the field enhancement effect is highly localized in the region that is very close to the corner for mode 3. While for mode 4, the field enhancement effect takes place in a larger volume near the corner. The LFE-LE resonance mode may be very useful for practical applications such as SERS and nonlinear optical enhancement.

Until now, we have figured out the physical origin of the five surface plasmon modes and attributed each mode to a particular class of plasmonic excitation mode. Next, we provide an effective way to manipulate the optical properties of silver nanoparticles which are important for practical applications. As CEP is a very crucial parameter which can dramatically affect almost all dynamical processes in laser–matter interaction, the ultrashort pulse provides a new scheme to regulate and control the LSPRs from the aspect of excitation optical field engineering. Figure 3 shows the influence of CEP on the charge distribution for a quadrupole mode at 447 nm and a dipole mode at 643 nm. The sign of the surface charge alternates between negative and positive as CEP is changed from $\varphi = 0.25\pi$ to $\varphi = 1.25\pi$. When $\varphi = 0.25\pi$ and $\varphi = 1.25\pi$, the silver nanocube reaches a maximum charge polarization for two modes. The response of the conduction electron cloud to the incident electric field is closely related with the optical field retardation effects. As CEP can dramatically affect the surface charge distribution, which directly determines the optical properties of silver nanoparticles, CEP becomes a powerful modulation tool on the LSPRs modes. This can be clearly seen from the calculation result as shown in Figure 4. When $\varphi = 0.75\pi$, only three plasmon modes are excited, which locate at 275 nm, 374 nm and 495 nm, respectively. The extinction cross sections of these three modes are much higher than those with $\varphi = 0.25\pi$. However, the mode at 643 nm is much weaker and the peculiar LFE-LE resonance mode at 447 nm is drastically suppressed. The reason is that the nanocube reaches a minimum charge polarization for two modes when $\varphi = 0.75\pi$ as shown in Figure 3. The suppression or promotion of the LFE-LE resonance mode can find potential applications in the area of SERS and nonlinear optical enhancement. Based on our discussion above, CEP can act as a powerful modulation means for the regulation and control of LSPRs, e.g., to implement selective plasmon mode excitation.

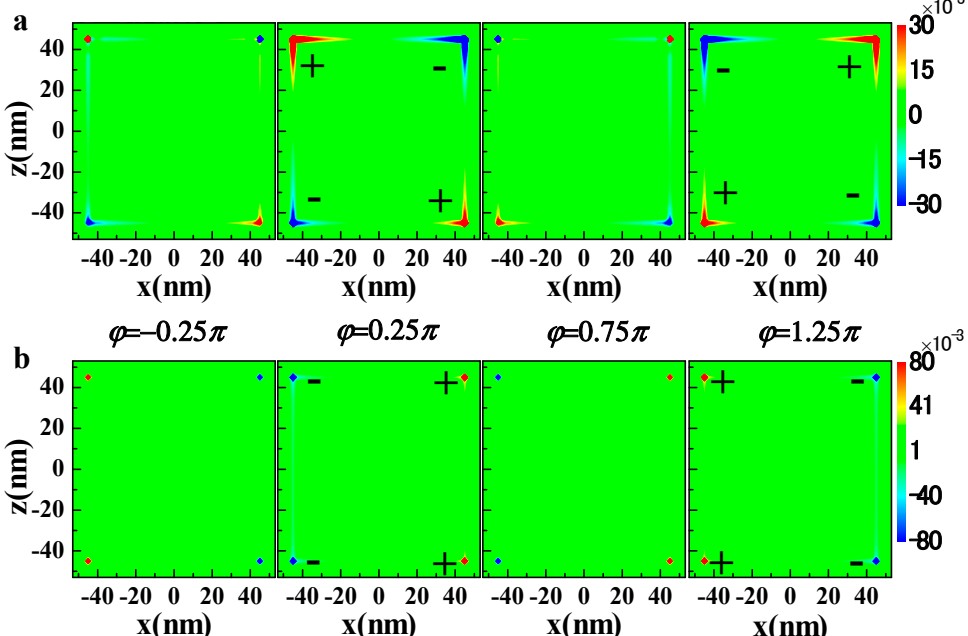

**Figure 3.** The influence of CEP on the surface charge distribution of a 90 nm silver nanocube. (**a**) Quadrupole mode at $\lambda = 447$ nm and (**b**) Dipole mode at $\lambda = 643$ nm with $\varphi = -0.25\pi$, $0.25\pi$, $0.75\pi$ and $1.25\pi$.

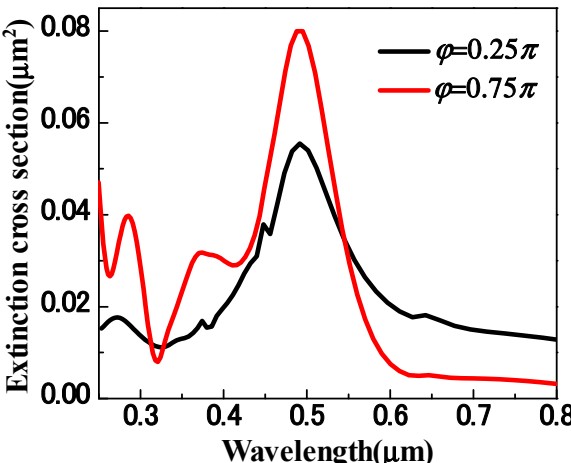

**Figure 4.** Comparison of the extinction cross section spectra for a 90 nm nanocubic silver particle with $\varphi = 0.25\pi$ and $\varphi = 0.75\pi$.

## 5. Conclusions

In summary, we have theoretically investigated the extinction spectra of a 90 nm silver nanocubic particle driven by a CEP-locked ultrashort pulse and found that five LSPR modes are excited by the incident waves. We have analyzed the physical origin of each mode from two aspects: the charge distribution on different parts of the cubic particle and the contributions of the dipole component and quadrupole component. The results show that the short-wavelength resonance mode mainly originates from the charges distributed at the faces of the cube, while the long-wavelength resonance mode mainly derives from the charges localized at the shape's corners. Charges located at edges contribute to the intermediate resonance modes located between the face mode and the corner mode, and other modes are generated by the cooperative contribution of charges at faces, edges and corners, together. On the other hand, the quadrupole mode occurs because of the onset of the electromagnetic retardation effect. From the two aspects, we have been able to specify unambiguously the physical origin of each LSPR peak in the extinction spectra.

We have presented an effective way to regulate and control the optical properties of nanoparticles by changing the CEP of the ultrashort excitation pulse. We have found that some plasmon modes can be drastically suppressed by changing CEP, especially the LFE-LE resonance mode. This suggests that CEP can act as a novel modulator of LSPR modes. As these LSPR modes are associated with very different optical properties regarding absorption, scattering, modal profile, local field enhancement, "hot spot" size and position and so on, the CEP could become a useful physical parameter to manipulate these properties for optimized applications in optical sensing, single-molecule detection, Raman spectroscopy, nonlinear optics, biomedical therapy and so on.

**Author Contributions:** Conceptualization, J.L. and Z.L.; methodology, Z.L.; software, J.L. and Z.L.; validation, J.L. and Z.L.; formal analysis, J.L. and Z.L.; investigation, J.L. and Z.L.; resources, Z.L.; data curation, J.L.; writing—original draft preparation, J.L.; writing—review and editing, Z.L.; visualization, J.L. and Z.L.; supervision, Z.L.; project administration, Z.L.; funding acquisition, J.L. and Z.L. All authors have read and agreed to the published version of the manuscript.

**Funding:** This research was funded by The National Natural Science Foundation of China, grant number 11904397.

**Institutional Review Board Statement:** Not applicable.

**Informed Consent Statement:** Not applicable.

**Data Availability Statement:** Not applicable.

**Conflicts of Interest:** The authors declare no conflict of interest.

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
