# Peer review of "Control of Surface Plasmon Resonance in Silver Nanocubes by CEP-Locked Laser Pulse"

_photonics, doi:10.3390/photonics9020053_

Round 1
Reviewer 1 Report
In the manuscript the authors present a method to control plasmon resonances in silver cubes by using an ultrashort carrier envelope phase locked laser pulse. The authors performed the calculations of the resonances under the excitation with the ultrafast laser conditions and proposed that by choosing the carrier envelope phase they can control the plasmon resonance. Although of potential interest to the reader of the journal the work misses some important information and requires improvement before it can be accepted for publication.
- Methods used in the work need to be better described. Modeling conditions are not presented. Approximations are missing, etc.
- Spectra discussed in figure 1 b and d are not really that different. Why do the authors claim the new methods provides a better way to excite LSPR?
- The figure 2 and relevant discussion of the modes deals with 2D cube, i.e. a square, not a 3D cube. As such some important information on the modes assignment is missing. It is possible that some of the modes discussed are of higher order which is not detected because of the limitations of the model considered. The authors must look at the 3D model.
- Finally, the authors need to consider the effects of polarization and excitation direction with respect to the cube orientation.
Author Response
Dear referee:
Thanks a lot for your review report on our manuscript. We would like to thank you for careful reading of the manuscript and comments and suggestions to improve the paper. We have taken into full account the useful comments and suggestions in the technical sides and made revisions accordingly, which will be described in the attachment.
Sincerely,
Ju Liu

Reviewer 2 Report
In this article, the authors have theoretically investigated the extinction spectra of a 90 nm silver nanocubic particle driven by a carrier-envelope phase (CEP) locked ultrashort pulse. They found that five localized surface plasmon resonance (LSPR) modes are excited by the incident waves. Some interesting conclusion was obtained based on the study. Authors reveal that the difference in the charge distribution on different parts of the cubic particle and the contributions of the dipole component and quadrupole component is the physical origin of the five plasmon modes. Then they modulate the LSPR spectra by changing CEP and the results show that CEP has selective plasmon mode excitation functionality and can act as a novel modulation role on LSPR modes. The modelling seems to be correct, the analysis is scientific and rigorous and the obtained results are physically sound. The reviewer thinks this manuscript should be published in Photonics in present form.
Author Response
Dear referee:
Thanks a lot for your careful reading of the manuscript and recommendation for publication.
Sincerely,
Ju Liu
Reviewer 3 Report
Liu and his coauthor investigated the plasmonic excitation of a 90-nm silver nanocubic particle in a CEP-lock ultrashort pulse by using FDTD simulation and found five LSPR modes and explained their physical origin. The results are interesting and provide useful guidance for plasmonic controlling that might be useful for practical application of plasmonic nanostructures. Therefore, I recommend for its publication. However, I suggest the authors further polish the manuscript before publication. The reasons are as follows:
- Please shorten the introduction section. Tthere are too much information which has been well reported, like”taka a spherical Ag nanoparticles as a good example……different modes would localize at different spatial regions of the nanoparticles.”
- Please summarize the main results at the final paragraph of the introduction section.
- Given that this manuscript is based on FDTD simulation, please detailed the simulation setup.
- Although the simulated results show five resonance peaks, many of them (e.g., peaks 2, 3, and 5) are very small. I am not sure they are real plasmonic resonance peak or just arise from calculation error, e.g., the mesh size is not small enough. Thus, please double check the calculation results, e.g., changing the mesh size to 1 nm or less.
Author Response

(The authors gave the same response as above.)

Round 2
Reviewer 1 Report
accept in present form
Reviewer 3 Report
I recommend for publicaiton as it is.